# A Comprehensive Molecular Characterization of the Pancreatic Neuroendocrine Tumor Cell Lines BON-1 and QGP-1

**DOI:** 10.3390/cancers12030691

**Published:** 2020-03-14

**Authors:** Kim B. Luley, Shauni B. Biedermann, Axel Künstner, Hauke Busch, Sören Franzenburg, Jörg Schrader, Patricia Grabowski, Ulrich F. Wellner, Tobias Keck, Georg Brabant, Sebastian M. Schmid, Hendrik Lehnert, Hendrik Ungefroren

**Affiliations:** 1Clinic for Hematology and Oncology, University Hospital Schleswig-Holstein, Campus Lübeck, D-23538 Lübeck, Germany; kim.luley@uksh.de; 2First Department of Medicine, University Hospital Schleswig-Holstein, Campus Lübeck, D-23538 Lübeck, Germany; bshauni@ymail.com (S.B.B.); Georg.Brabant@manchester.ac.uk (G.B.); Hendrik.Lehnert@uni-luebeck.de (H.L.); 3Group for Medical Systems Biology, Lübeck Institute of Experimental Dermatology, University of Lübeck, D-23538 Lübeck, Germany; Axel.Kuenstner@uni-luebeck.de (A.K.); hauke.busch@uni-luebeck.de (H.B.); 4Institute for Cardiogenetics, University of Lübeck, D-23538 Lübeck, Germany; 5Institute for Clinical Molecular Biology, University of Kiel, Kiel, D-24105 Kiel, Germany; s.franzenburg@IKMB.uni-kiel.de; 6First Medical Department—Gastroenterology and Hepatology, University Medical Center Hamburg-Eppendorf, D-20246 Hamburg, Germany; jschrader@uke.de; 7Department of Medical Immunology, Charité Berlin, Corporate Member of Freie Universität Berlin, Humboldt-Universität zu Berlin and Berlin Institute of Health, D-13353 Berlin, Germany; patricia.grabowski@charite.de; 8Department of Surgery, University Medical Center Schleswig-Holstein, Campus Lübeck, D-23538 Lübeck, Germany; Ulrich.Wellner@uksh.de (U.F.W.); tobias.keck@uksh.de (T.K.); 9Institute for Endocrinology and Diabetes, University of Lübeck, D-23538 Lübeck, Germany; Sebastian.Schmid@uni-luebeck.de; 10University of Salzburg, A-5020 Salzburg, Austria; 11Clinic for General Surgery, Visceral, Thoracic, Transplantation and Pediatric Surgery, University Hospital Schleswig-Holstein, Campus Kiel, D-24105 Kiel, Germany

**Keywords:** pancreatic neuroendocrine tumor, BON-1, QGP-1, NT-3, β-cell, EMT, microRNA

## Abstract

Experimental models of neuroendocrine tumor disease are scarce, with only a few existing neuroendocrine tumor cell lines of pancreatic origin (panNET). Their molecular characterization has so far focused on the neuroendocrine phenotype and cancer-related mutations, while a transcription-based assessment of their developmental origin and malignant potential is lacking. In this study, we performed immunoblotting and qPCR analysis of neuroendocrine, epithelial, developmental endocrine-related genes as well as next-generation sequencing (NGS) analysis of microRNAs (miRs) on three panNET cell lines, BON-1, QGP-1, and NT-3. All three lines displayed a neuroendocrine and epithelial phenotype; however, while insulinoma-derived NT-3 cells preferentially expressed markers of mature functional pancreatic β-cells (i.e., *INS*, *MAFA*), both BON-1 and QGP-1 displayed high expression of genes associated with immature or non-functional β/δ-cells genes (i.e., *NEUROG3*), or pancreatic endocrine progenitors (i.e., *FOXA2*). NGS-based identification of miRs in BON-1 and QGP-1 cells revealed the presence of all six members of the miR-17–92 cluster, which have been implicated in β-cell function and differentiation, but also have roles in cancer being both oncogenic or tumor suppressive. Notably, both BON-1 and QGP-1 cells expressed several miRs known to be negatively associated with epithelial–mesenchymal transition, invasion or metastasis. Moreover, both cell lines failed to exhibit migratory activity in vitro. Taken together, NT-3 cells resemble mature functional β-cells, while both BON-1 and QGP-1 are more similar to immature/non-functional pancreatic β/δ-cells or pancreatic endocrine progenitors. Based on the recent identification of three transcriptional subtypes in panNETs, NT-3 cells resemble the “islet/insulinoma tumors” (IT) subtype, while BON-1 and QGP-1 cells were tentatively classified as “metastasis-like/primary” (MLP). Our results provide a comprehensive characterization of three panNET cell lines and demonstrate their relevance as neuroendocrine tumor models.

## 1. Introduction

Preclinical models of neuroendocrine tumors (NET) currently encompass only a limited number of human NET cell lines and mouse models. The most commonly used human neuroendocrine neoplastic cell lines of gastroenteropancreatic (GEP)-NET origin encompass the pancreatic NET (panNET) cell lines BON-1 (BON) [1], QGP-1 (QGP) [2,3], and CM [4] as well as the small intestinal NET (siNET) cell lines GOT-1, KRJ-I, P-STS, H-STS, and L-STS. However, GEP-NET cell lines have only partially been characterized and, hence, several issues still remain open: i) Their authenticity with regard to their neuroendocrine origin and phenotype, ii) their genomic and mutational characteristics, and iii) the identity of their normal counterparts from which they were originally derived. Recently, in the course of analyzing a panel of siNET and panNET cell lines by exome sequencing and genome-wide copy number, it was revealed that the KRJ-I, H-STS, and L-STS lines did not exhibit a neuroendocrine phenotype and are in fact lymphoblastoid cells [5]. For other cell lines, their mutation patterns, and proliferation rates seem distinct from well-differentiated NETs in patients [5,6,7] and thus might not adequately reflect the tumor biology of well-differentiated NETs. Specifically, for panNET, studies in BON and QGP cells have raised questions regarding their relevance as models due to the absence of mutations in *MEN1* [5,6,7] or low SSTR expression [8]. Together, this emphasizes the need for careful reevaluation and further characterization of the existing cell lines. One aim of the present study was; therefore, to confirm the authenticity of the BON and QGP cell lines with respect to their neuroendocrine and epithelial phenotype, developmental origin, and propensity for cell motility in vitro.

Although BON and QGP cells have been compared for the expression of some classical neuroendocrine markers to a recently established patient-derived panNET cell line (NT-3, [9]) and to panNET tissues [8], such a comparison has not yet been performed for other cellular features such as epithelial/mesenchymal differentiation, expression of genes governing immature and mature β-cell function and differentiation from pancreatic (endocrine) progenitors, or microRNA (miR) signatures. Earlier, we have performed miR profiling in GEP-NET and panNET tissues [10,11]; however, tumor tissues are heterogeneous with respect to cellular composition and, hence, their analysis does not allow for the identification of miRs expressed specifically by the tumor cell fraction.

Recently, a cross-species analysis has revealed the existence of previously unrecognized subtypes of panNET in both mice and humans, and could assign different mutations and phenotypic, clinical, and pathologic properties to these tumor subtypes underlying the heterogeneous biology of this disease. Specifically, dual mRNA and miR transcriptome profiling analysis has identified three distinct molecular subtypes and associated biomarkers in human panNET, termed “islet/insulinoma tumors” (IT), “metastasis-like/primary” (MLP), and “intermediate” [12]. PanNETs of the IT subtype consist primarily of less-aggressive, non-metastatic insulinomas that expressed genes associated with insulinomas and differentiated/mature β-cells. In contrast, tumors of the MLP subtype are invasive/metastatic and their signatures are enriched for genes associated with immature non-functional β-cells, and eventually EMT, fibroblasts/stroma, and stem cells, implicating a progenitor origin. The intermediate subtype includes mostly nonfunctional panNETs, shares many genes with the IT subtype, and is moderately associated with metastasis. An association of the newly-defined transcriptional subtypes with the WHO classification of NET grades showed that G1 and G2 human panNETs are heterogeneous, variably associating with all three transcriptome subtypes, whereas high-grade NET G3 tumors are exclusively associated with the MLP subtype [12].

Based on results from other studies we postulate that BON and QGP cells possess, at least partially, a neuroendocrine and well-differentiated epithelial phenotype associated with a low invasive potential. However, since both lines classify as tumor cells they might have undergone a dedifferentiation process or, alternatively, have turned malignant already at an early developmental stage. In this case these cells should resemble immature islet cells or pancreatic precursors. To analyze this in more detail, we have carried out a comprehensive phenotypic characterization of the BON and QGP cell lines with respect to their differentiation and developmental states by protein, mRNA and miR expression analyses as well as to their invasive potential by assessing the cells’ migratory ability in vitro. In addition, attempted an allocation of both cell lines to one of the above mentioned molecular subtypes of panNETs.

## 2. Results

### 2.1. Expression of Markers of Neuroendocrine Differentiation

Initially, we evaluated the extent of neuroendocrine differentiation of BON and QGP cells by measuring the expression of a panel of neuroendocrine markers using quantitative real-time RT-PCR (qPCR) and immunoblot analysis. A primary panNET cell line, NT-3, recently characterized by us [9], was used as control. We found, in Western blot analysis, strong signals for Synaptophysin (SYP) in NT-3 cells and weaker ones in BON and QGP cells (Figure 1A, upper blot). The expression of Chromogranin A (CgA, encoded by *CHGA*) was quite similar at both the mRNA and protein level, except that no specific signals were detectable for QGP in the immunoblots (Figure 1B). 

In BON, QGP, and NT-3 cells we also evaluated expression of somatostatin receptor 2 (SSTR2) by both qPCR and immunoblotting. At the mRNA level, SSTR2 expression in BON was 2% and in QGP 15% of the levels in NT-3 (Figure 1C), whereas signals for SSTR2 in the immunoblots were comparatively weak in NT-3 cells and hardly detectable in BON and QGP cells (Figure 1C). The pattern of SSTR5 mRNA expression was similar to that of SSTR2 with levels of BON and QGP being 6% and 32%, respectively, that of NT-3 (Figure 1D). Additional markers of neuroendocrine differentiation included Neural cell adhesion molecule 1 (NCAM1, encoded by *NCAM1*) which shows equal expression in QGP and NT-3 whereas that in BON cells was ~30-fold lower (Figure 1D). The transcript levels of Neurofilament light polypeptide (NFL, also termed Neurofilament 68, NF68, encoded by *NEFL*), a member of the intermediate filament family, were 5.7-fold lower in BON and 26,000-fold lower in QGP when compared to NT-3 cells (Figure 1D). Finally, expression of Microtubule-associated protein 2 (MAP2, encoded by *MAP2*) was 5-fold higher in QGP but 500-fold lower in BON than in NT-3 cells (Figure 1D). These results confirm the neuroendocrine phenotype of BON and QGP cells albeit their marker expression was generally lower than in NT-3 cells. 

### 2.2. Expression of Markers of Epithelial/Mesenchymal Differentiation

The panNETs are likely derived from pancreatic ducts [13] and; therefore, should have an epithelial phenotype. To confirm this, we assessed expression of the epithelial marker E-cadherin in BON and QGP cells by immunoblotting. BON, QGP, and NT-3 cells were all strongly positive for E-cadherin and protein levels were higher than in the two *CDH1*^low^ PDAC-derived cell lines Panc1 and MiaPaCa2 (Figure 2A). To independently confirm the epithelial phenotype, we probed the same blot with an antibody to the mesenchymal marker Vimentin, which should in this case result in low or absent signals. To this end, Vimentin protein levels were low in BON and NT-3 and absent from QGP cells (Figure 2B). This expression profile of E-cadherin/Vimentin in BON and QGP cells is indicative of a well-differentiated epithelial phenotype and contrasts with the quasimesenchymal phenotype (*CDH1*^low^/*VIM*^high^) of the Panc1 and MiaPaCa2 lines [14]. 

### 2.3. Expression of Genes of Islet Cell Hormones and Mature β-Cell Differentiation

As mentioned above, BON and QGP malignant conversion might have occurred at an early developmental stage, in which case these cells should resemble immature rather than functional islet cells, and eventually share some traits in common with pancreatic precursors. To verify or dismiss such a scenario, we evaluated in BON, QGP, and NT-3 cells expression of Insulin, Glucagon, and SST (encoded by *INS1*, *GCG*, and *SST*, respectively) by qPCR analysis. Consistent with its derivation from an insulinoma, *INS1* was found to be highly expressed and secreted by NT-3 cells [9]. Here, *INS* mRNA was present in all three cell lines with expression in NT-3 being ~5-fold greater than in QGP and ~100-fold greater than in BON cells (Figure 3). Likewise, *GCG* mRNA was 250–500-fold more abundant in NT-3 vs. BON and QGP (Figure 3). Interestingly, we detected extremely high mRNA levels for SST in QGP cells (~100-fold higher than in NT-3 and ~1000-fold higher than in BON cells, Figure 3).

Next, we measured in BON, QGP, and NT-3 cells expression of *MAFA*, *PDX1*, and *ACVR1C,* which are all mature β-cell-specific factors. *MAFA*, a marker gene of mature functional β-cells and key regulator of glucose-stimulated insulin secretion, was expressed at a much higher level in NT-3 than in BON (~200-fold less) and QGP (~10-fold less) cells (Figure 3). PDX1 (also termed Insulin promoter factor 1, IPF1) is a transcription factor (TF) that specifies the development of mono-hormonal insulin-secreting cells and poly-hormonal pancreatic β-cells. The *PDX1* gene is upregulated at later stages of β-cell development, is restricted to β-cells in adult pancreas, and controls the expression of *MAFA* and *INS1*. Its mRNA expression in NT-3 was ~10-fold higher than in BON and ~50-fold higher than in QGP cells (Figure 3). The mRNA for activin receptor-like kinase 7 (ALK7, encoded by *ACVR1C*), a receptor regulating insulin secretion and β-cell function in response to Nodal or Activin B binding [15], was most abundant in NT-3 cells, being ~5-fold higher than in BON and ~10-fold higher than in QGP cells (Figure 3). Taken together, when compared to NT-3 cells, BON and QGP cells are characterized by much lower expression of markers of β-cell function and differentiation.

### 2.4. Expression of Genes of Immature β-Cells and Lineage-Specific Pancreatic Progenitors

Having shown that BON and QGP cells are largely deficient in mature β-cell-specific gene expression, we pursued the idea that they resemble immature and non-functional pancreatic β-cells, or endocrine progenitors. To test this possibility, we transcriptionally profiled BON and QGP cells using a battery of genes involved in pancreatic, pancreatic endocrine, and islet cell differentiation. PAX4 which is crucial for β- and δ-cell development, β/δ-cell lineage bifurcation, and conversion of α- or δ-cells into functional β-like cells [16] is restricted at later stages to β-cells. We found that the transcript levels for PAX4 are 10–20-fold more abundant in NT-3 cells than in the two permanent cell lines (Figure 4A). Similarly, the related PAX6, which is expressed in all endocrine cells and increases the number of islet cells, was ~10-fold higher in NT-3 than in QGP and ~1500-fold higher than in BON cells (Figure 4A).

*NEUROD1* is a developmentally regulated gene that continues to be expressed in mature β-cells and is essential for their maturation by assembling the insulin exocytotic machinery and driving *INS* expression, and by downregulating genes that impair insulin secretion in β-cells. Interestingly, *NEUROD1* mRNA is 5-fold more abundant in BON than in NT-3 cells (Figure 4B). Likewise, expression of Neurogenin3 (NGN3, encoded by *NEUROG3*) was ~4-fold higher in QGP compared to NT-3 cells (Figure 4B). NGN3, a marker for pancreatic endocrine progenitors and activator of TFs responsible for differentiation of the endocrine precursors into mono-hormonal islet cells, disappears during the final stage of β-cell differentiation [17].

ISL1, a TF binding to *INS* enhancer sequences, is 43-fold lower in BON compared to NT-3 cells (Figure 4C). In contrast, CDX2 which by synergistic interactions with PAX6 can activate gene transcription of GCG (Figure 4C), GATA4, a TF expressed during early pancreatic budding, and GATA6 were either equally expressed in BON, QGP and NT-3 (CDX2, GATA4) or were even higher in BON and QGP compared to NT-3 cells (GATA6). The mRNA abundance of NKX2.2, a TF important for development of α-, β-, and δ-cells and for maintenance of β-cell identity, is 3-fold lower in BON and 30-fold lower in QGP than in NT-3 cells (Figure 4E), while NKX6.1, which is crucial for islet endocrine cell development and insulin secretion in β-cells, is 3-fold lower in QGP and 1500-fold lower in BON cells (Figure 4E). 

HNF3β/FOXA2 is a TF expressed in foregut endoderm which differentiates into pancreatic buds. Its expression continues in all pancreatic cell types and during pancreatic development and peaks in pancreatic islets where it is important for the development and functionality of β-cells. Interestingly, FOXA2 transcript levels are 3–4-fold higher in both BON and QGP compared to NT-3 cells (Figure 4F). *PTF1A*, a gene that plays a role in determining whether cells allocated to the pancreatic buds continue towards pancreatic organogenesis or revert to duodenal fates, has its strongest expression in QGP cells, being 70-fold higher and in BON 6.5-fold higher than in NT-3 cells (Figure 4F). The data from the developmental gene expression survey clearly show that BON and QGP cells have higher expression of genes involved in pancreatic and early but not late endocrine differentiation of the pancreas. 

### 2.5. Identification of MiRs Expressed in BON and QGP Cells

MiRs are involved in various aspects of cellular function, normal pancreas development, islet differentiation, or pancreatic tumorigenesis [18], and represent potential prognostic markers in NET [19]. For identification of miRs we subjected total RNA samples of untreated BON, QGP, and NT-3 cells to next-generation sequencing (NGS) analysis. We were able to identify a total of 43 miRs in BON and 50 miRs in QGP cells, of which 15 and 12 miRs, respectively, were novel and previously unknown (Appendix A). Several of the known miRs expressed in BON and QGP cells (Let7F1, 7–3, 17, 19B1, 20A, 33A, 182, 221, 664b, 1225, 1248, 3609, 3653) as well as some novel miRs were moderately or even highly abundant with relative expression scores (RESs) ranging from 0 to 12,533. Some miRs were selectively expressed only in BON (15a, 15b, 126, 129–1, 431, 503, 3618, 3913–2, 6516, 6820, 6895) or QGP (7–1, 17, 18a, 19a, 21, 29b2, 92a1, 98, 146a, 196A2, 483, 671, 3131, 3691, 6789, LET7B, LET7F1) cells, or were shared by both lines (7–3, 17, 19b1, 20a, 33a, 141, 182, 221, 664B, 1225, 1248, 3609, 3653) (Appendix A). 

We hypothesized that the miR signatures of BON, QGP, and NT-3 cells should exhibit a partial overlap in miRs involved in differentiation and development. In contrast, miRs regulating mature β-cell function should be enriched in NT-3 cells, while those associated with malignancy or response to drug treatment should predominate in BON and QGP cells. To this end, all three cell lines, to a various extent, express the six members of the miR-17–92 cluster (17, 18a, 19a, 19b-1, 20a, 92a), a highly-conserved gene cluster with various roles in β-cell function and differentiation [20,21,22,23,24]. Yet other miRs involved in β-cell/islet cell development and insulin secretion (7–3, 15a, 33a, 182, and 375 [25,26,27,28,29,30]) are more prominent in NT-3 than in BON and QGP cells. For instance, miR-375, which was only detected in NT-3 but not in BON or QGP cells, is one of the most abundant miRs in the islets and has multiple functions, such as control of β-cell identity and β-cell mass as well as regulation of insulin secretion [31]. In fact, the malignant nature of the BON and QGP cells was evident from the expression of miRs for which oncogenic or anti-oncogenic effects have been reported (i.e., the miR-17–92 cluster that also exhibits both oncogenic and tumor suppressive properties [32,33] and regulates multiple aspects of pancreatic tumor development and progression [34,35]). Another miR, miR-3653, is particularly interesting due to its abundance in both BON and QGP cells and its negative association with tumor size, lymph node metastasis, and poor survival in glioma [36], its metastasis and EMT suppressor function in colon cancer cells [37], and its upregulation in panNET patients with distant metastases [38]. Taken together, it appears that miRs involved in mature β-cell function are preferentially expressed in NT-3 cells and lower or absent in BON and QGP. This; however, does not apply for members of the miR-17–92 cluster, which are likely to exert tumor-associated functions in BON and QGP cells. For a list of the more abundantly expressed miRs and their functional activities see Appendix A.

### 2.6. Migratory Activity of BON and QGP Cells

The neuroendocrine and well-differentiated epithelial phenotype of BON and QGP cells described above is suggestive of a low invasive potential. To test this prediction more directly we measured random and directed migratory activity in a chemokinesis and a chemotaxis setup, respectively, in real-time on an xCELLigence platform. In the chemotaxis setup, cells are attracted by the high-serum content in the lower chamber and, if motile, move through the membrane pores towards the lower chamber. We observed that under chemokinesis conditions BON cells exhibited a rapid and transient but comparatively weak migratory response (Figure 5A, QGP cells were not tested). However, when the assay is run with a chemotaxis setup with 10% fetal bovine serum (FBS) as attractant, neither BON nor QGP cells showed any detectable migratory activity (regardless of whether collagen I or collagen IV was used for coating), in contrast to highly invasive Panc1 cells which were used as a positive control [39] (Figure 5B). It remains to be seen; however, if certain cytokines or growth factors can stimulate migration or invasion in BON and QGP cells.

## 3. Discussion 

The goal of the present study was to confirm the authenticity of the BON and QGP cell lines and to better characterize their state of epithelial and pancreatic endocrine differentiation. To assess the neuroendocrine phenotype of our BON and QGP1 cultures, we analyzed expression of a panel of classical neuroendocrine markers in comparison to a lymph node-derived cell line (NT-3) from a male patient with well-differentiated panNET. In both BON and QGP, SYP and, to a lesser extent, CgA protein was readily detectable by immunoblotting but was clearly lower than in NT-3 cells. This is in line with results from an immunohistochemical study reporting CgA protein to be weak in BON and absent in QGP cells [5]. CgA has been shown to be a reliable serum diagnostic biomarker for panNETs but not for insulinomas [40]. Both BON and QGP cells were positive for SSTR2 and -5 mRNA, although levels were 10–20-fold lower compared to NT-3. Exner and coworkers assessed SSTR mRNA expression in BON and QGP1 cells in comparison to panNET and siNET tissues, and observed that while the abundance of SSTR2 mRNA in BON and QGP cells was approximately 100-fold lower, that of SSTR5 mRNA in BON cells was close to that in NET tissues [8]. From these findings and additional expression data of other neuroendocrine markers (NCAM1, NF68, MAP2) we conclude, in agreement with others [5], that both BON and QGP cells possess a neuroendocrine phenotype.

Regarding the expression of EMT markers in BON and QGP we detected, by immunoblotting, strong expression of E-cadherin in BON, QGP, and NT-3 cells. Readily-detectable levels of E-cadherin in BON cells have also been reported by Wu and colleagues [41]. The epithelial differentiation of these cell lines was confirmed by low protein expression of the mesenchymal marker Vimentin as compared to the poorly differentiated and *VIM*^high^ PDAC cell lines, Panc1 and MiaPaCa2 [42]. The epithelial phenotype may, in addition, be maintained by high expression of miR-3653 [38], which has been shown to inhibit mesenchymal transition by targeting the EMT-associated TF Zeb2. In this context, it is noteworthy that members of the miR-200 family, whose members are strong inducers of epithelial differentiation and are repressed by Zeb1 [43], were not among the miRs identified in BON, QGP, and NT-3 cells. 

Our qPCR-based expression analysis revealed that mRNA levels of *INS* were much lower in BON and QGP than in NT-3 cells. The high abundance of *INS* mRNA in NT-3 cells is in line with detection of the protein in NT-3 culture supernatants [9]. Transcript levels of other genes implicated in regulation of insulin synthesis and secretion were also lower in BON/QGP (i.e., *PDX1*, *ISL1*, *MAFA,* and *ACVR1C*). The comparatively low expression in BON and QGP of ALK7, a receptor from the TGF-β/activin receptor superfamily that suppresses ectopic proliferation during panNET development [15], may contribute to the higher Ki67 index observed in both cell lines compared to panNET tissues and primary cell cultures [9]. Finally, mRNA levels of *NKX2.2*, a TF important for maintenance of β-cell identity, and *NKX6.1*, a related TF important for islet endocrine cell development and insulin secretion in β-cells, were lower in BON and QGP compared to NT-3 cells. Regarding markers for other hormone-producing cell types, we found *GCG* mRNA levels in BON and QGP cells to be two orders of magnitudes lower than in NT-3 cells. Of note, expression of PAX6, which can activate transcription of GCG, is also lower in BON and QGP relative to NT-3 cells. *SST* mRNA was dramatically increased in QGP over that in NT-3 and BON cells, which is in agreement with the establishment of the QGP cell line from a SST-producing islet cell carcinoma [3]. 

Remarkably, some marker genes of immature, non-functional β-cells (i.e., *NEUROD1*), and of pancreatic and pancreatic-endocrine progenitors not expressed during the final stage of β-cell differentiation (i.e., *NEUROG3*), stand out in that their mRNA levels were more abundant in QGP and BON than in NT-3 cells. It should be noted that NEUROG3 activates *NEUROD1*, which could help to explain the comparatively high expression of *NEUROD1* in BON and QGP cells. Yet other TFs-encoding genes expressed in foregut endoderm and during early pancreatic budding (*GATA4*, *GATA6*, *FOXA2*, *PTF1A*) were either expressed at equal levels among BON, QGP, and NT-3 cells (differences <2-fold), or were 3–4-fold (*GATA6, FOXA2*) or even up to 70-fold (*PTF1A*) more abundant in BON and QGP compared to NT-3 cells. The high *PTF1A* mRNA levels may be the result of absent (BON) or low (QGP) levels of miR-18a, which regulates expression of Ptf1a in pancreatic progenitors. Together, the results from gene expression analyses of islet cell hormones and mature β-cell differentiation markers provide evidence for the hypothesis that BON and QGP cells resemble immature β-cells or pancreatic precursors, rather than mature functional β-cells. In contrast, in all panNETs analyzed by Cejas and coworkers the mature cell transcriptional signature was stronger than that of progenitors [17].

The NGS-based identification of miRs expressed in BON and QGP cells yielded a large number of miRs with potential involvement in panNET biology (see Appendix A). The absent or only low expression in BON or QGP compared to NT-3 cells of miR-15a/b, miR-375, and miR-182, which control β-cell identity, β-cell mass, and upregulate *INS* expression and secretion [27,29,31], attests to the immature β-cell/progenitor phenotype of BON and non-β-cell identity of QGP cells, respectively. This is further strengthened by the absence in BON or QGP cells of yet other miRs, for which a role in endocrine differentiation or β-cell function has been reported (23a, 27b-3p, 29c, 34a, 96, 103, 107, 124, 125b-5p, 130a/b, 152, 184, 192–5p, 195, 212/132, 335, 342, 376b-3p, 451) [18,44,45,46]. In summary, BON and QGP cells express only few miRs specific to insulinomas or mature/functional β-cells and, if present, transcript levels are low, which is consistent with the lack of *INS* expression in both lines. High levels of miR-196a, miR-21, and miR-221 have been reported to be associated with tumor proliferation, advanced stage, metastases, recurrence, or reduced survival [11,47] and mucinous precursor lesions of PDAC [48]. Moreover, miR-664b identified in both BON and QGP cells regulates multiple aspects of pancreatic tumor development and progression and directly targets *PAX6,* which might help to explain the low expression of *PAX6* in BON and QGP cells (see Figure 4A). Like panNET, QGP cells exhibit coordinate expression of miRs 17, 20, and 92, which are oncogenic in association with *MYC* [49]. Generally, some of the tumor-associated miRs might contribute to the higher proliferation index of BON and QGP compared to panNETs or authentic NET cell lines like miR-18a. This miR counteracts EGFR and AKT to inhibit the proliferation of pancreatic progenitors [50] and might explain the poor sensitivity of these cells to the mTOR inhibitor RAD001 (everolimus) [51]. This is of note given that everolimus is being used for the treatment of patients with panNETs [51,52]. Interestingly, inhibition of mTOR by everolimus has been shown to result in activation of Src in panNET cells [53]. Since Src is crucial for their invasive and metastatic activities [54], the use of rapalogues could increase the migratory capacity of BON and QGP cells on different scaffolds but in a therapeutic setting bears a potential risk of serious side effects.

Our NGS-based analysis of miRs also allowed for the identification of previously unknown miRs. The novel miRs identified here (15 in BON and 12 in QGP), some of them with extremely high RES (up to 12,533), might be useful in further refining the subtype classification of BON and QGP cells following elucidation of their biological function.

Based on the mRNA expression patterns and the identified miRs, we attempted to allocate BON and QGP cells to one of the three transcriptional subtypes described by Sadanandam and colleagues [12]. Both BON and QGP cells can tentatively be associated with the MLP subtype, while NT-3 cells resemble the IT subtype (Figure 6). This conclusion is based on the finding that BON and QGP have generally lower expression of markers of a neuroendocrine phenotype (SYP, CgA), but higher expression of pancreatic/islet cell precursor genes that are not expressed during the final stage of β-cell differentiation (i.e., *NEUROG3*). Conversely, NT-3 cells have generally much lower, although readily detectable, mRNA levels of pancreatic precursor genes. Most importantly, BON and QGP fail to express significant mRNA levels of genes typical of mature/functional β-cell differentiation (*INS1*, *MAFA*, *ISL1*, *PDX1*, *ACVR1C*) or in maintaining mature β-cell function and identity (*MAFA*, *PDX1*, *NKX2.2*, *NKX6.1*). All these genes are poorly expressed in MLP tumors, but high in IT tumors [12] and in NT-3 cells [9]. In the above mentioned study, a 30-miR human panNET signature was clustered into three distinct subsets (miR-cluster-1, -2, and -3), and each of these three miR-clusters was enriched in one of the three mRNA transcriptome subtypes [12]. When we compared this 30-miR signature with the miRs identified in BON and QGP cells, we found six miRs that were shared by QGP (7–1, 17, 20a, 92a, 221) and BON (17, 20a, 126, 221) cells. However, except for miRs 17 and 20a, the abundance was too low to be of biological significance. Taken together, the data from both the developmental mRNA survey and the overall miR expression pattern suggest that BON and QGP cells are unlikely to belong to the IT transcriptomic subtype but, rather, resemble the MLP subtype (Figure 6). 

Within the MLP subtype of panNETs one half of primary tumor samples was poorly metastatic (termed MLP-1) while the other half (termed MLP-2) had high metastatic potential [12]. Our results on the low-to-absent migratory activity of BON and QGP cells in vitro is in line with high expression of the invasion suppressor E-cadherin in BON [41,55] and QGP cells, and low or undetectable expression of Vimentin, together predicting a low invasive/metastatic capacity. In orthotopic mouse models, wild-type BON cells showed a variable and comparatively low metastatic spread to the liver with metastases found in 0% [56], 12.5% [41], or ~50% [57] of transplanted mice, while equivalent data are not available for QGP1. This is also supported by the miR data in three ways. i) Expression in BON and QGP of miRs that inhibit migration, invasion, metastasis, and EMT [35,37,38,58,59]; ii) the identification in QGP but not BON cells of only two, weakly expressed miRs associated with metastases of panNET, miR-21 (RES = 9) and miR-let7b (RES = 6); and iii) the absence in the BON and QGP lines of mouse MLP and metastasis-specific miRs highly expressed in the majority of the samples in miR-cluster-1 and -2 from human panNET [12,60], and of miRs previously shown by us to be associated with NET metastases [11]. Together with the observation that BON and QGP cells display only low migratory in vitro and low-to-moderate metastatic activity in vivo, both lines are; therefore, likely to belong to the MLP-1 sub-subtype, where 5% of the tumors were associated with distant metastases vs. 45% of the MLP-2 subtype [12] (Figure 6). However, Sadanandam and coworkers identified high *VIM* expression in MLPs compared with ITs, although no data were presented as to whether this applies to both MLP-1 and MLP-2 tumors. Since BON and QGP cells are unlikely to have undergone an EMT—both lines are *CDH1*^high^/*VIM*^low^ and abundantly express miR-3653, which suppresses metastasis and EMT [37,38]—they might represent yet another (still hypothetical) subtype that combines features of both MLP and IT (designated “IT/MLP-1 Intermediate” in Figure 6). Intriguingly, Sadanandam et al. found six human MLP tumors that clustered more closely than most MLPs with human and mouse ITs, and these were suspected to represent a subset of human MLPs with increased IT-specific genes [12]. In addition, they described another three human panNETs that were positive for both insulin and ENPP2 (designated Ins-hi MLP subtype) and thus combined features of MLP tumors (*ENPP2*^high^ and *INS*^low^) and IT tumors (*ENPP2*^low^ and *INS*^high^). This led these authors to suggest that ENPP2 along with insulin may serve as a biomarker that can be used to distinguish not only the IT and MLP subtypes, but also sub-subtypes of MLPs [12]. It will be interesting to see if the subtype classification of the BON and QGP cell lines can be further refined with the use of ENPP2, and insulin/serotonin [1,61] or SST [3], respectively.

The expression of EMT genes in MLPs suggested that a dedifferentiation process has occurred in a subset of tumors that progressed from IT into metastatic MLP tumors [12], revealing the existence of an early and separate cell or pathway of origin for prometastatic MLP-2 tumors. In contrast, MLP tumors with an epithelial phenotype (*VIM*^low^, *CDH1*^high^) and Ins-hi, or by analogy SST-hi, arise either directly from mature pancreatic β- or δ-cells or via progression of β- or δ-cell–derived IT tumors. Such a scenario may hold true for QGP cells given the high SST expression and the relatively high co-expression of *PAX4* and *PAX6*.

As a word of caution it should be mentioned that the BON and QGP lines harbor mutations in tumor suppressor and oncogenes that do not normally occur in panNETs, and may have been acquired at some point during in vitro culture as a result of selective pressure (i.e., *TSP53* [5,7], *CDKN2A/B,* and *DPC4* [5,62]). Given the potential impact of these cancer-associated mutations on various aspects of cellular and neuroendocrine function, one should be aware that data derived from therapeutic interventions in these cell lines may not be predictive of the clinical response in panNET patients. 

## 4. Conclusions

In this study, we showed that the BON and QGP cell lines display a neuroendocrine and well-differentiated phenotype, but, in contrast to the NT-3 line, do not exhibit a gene signature characteristic of mature/functional pancreatic islet/β-cells. Rather, they preferentially express genes normally activated during pancreatic development or in early endocrine progenitors, compatible with arrested differentiation. The absence of mesenchymal traits, the expression of microRNAs involved in suppression of invasion and metastasis, as well as the low migratory and metastatic activity further suggest that BON-1 and QGP-1 cells have arisen either from mature islet cells or via progression of β- or δ-cell–derived tumors of the IT transcriptional subtype, rather than by dedifferentiation of IT tumors into tumors with a MLP-2 transcriptional subtype. We conclude that both cell lines are principally relevant as neuroendocrine tumor models and that the results of this study could aid in deciding whether or not the use of these cell lines is feasible for novel study objectives.

## 5. Material and Methods

### 5.1. Cells

The BON cell line was established from a functional human pancreatic carcinoid tumor of the pancreas [1] and was provided by C.M. Townsend (University of Texas, Galveston, TX, USA). BON cells are functional with respect to secretion of serotonin [1,61], a monoamine that is released from a subset of pancreatic islet cells and β-cells along with insulin in a glucose-dependent manner [63]. The QGP-1 line was generated from a primary pancreatic carcinoma [2] and was purchased from the JCRB Cell Bank (Osaka, Japan). Cells were cultured in DMEM/Ham’s F12 (1:1) (BON), or RPMI 1640 (QGP), supplemented with 10% FBS and penicillin/streptomycin. The NT-3 cell line was established and characterized by us recently [9]. NT-3 cells were maintained under semi-adherent conditions in collagen IV-coated culture flasks in RPMI 1640 medium supplemented with 10% FBS, penicillin/streptomycin, HEPES, EGF (20 ng/mL), and FGF2 (10 ng/mL) (Peprotech, Hamburg, Germany).

Human WI-38 fibroblasts were included as negative control in the qPCR assays, while the pancreatic ductal adenocarcinoma (PDAC) cell lines Capan2, Panc1, and MiaPaCa2 were used as controls for expression of E-cadherin (encoded by *CDH1*) or Vimentin (encoded by *VIM*) [42]. Since Panc1 cells are highly invasive, these cells were used along with another non-motile PDAC cell line, Colo357, as controls in the migration assays. Panc1 cells are of ductal/exocrine origin; however, they possess some neuroendocrine features, such as expression of CgA and SSTR2 [64] and the potential to be transdifferentiated into insulin-producing cells [65,66,67,68].

### 5.2. RNA Isolation and Quantitative Real-Time RT-PCR (qPCR)

RNA isolation and purification for qPCR was done with PeqGold (PeqLab, Erlangen, Germany). Total RNA (2.5 μg) was reverse transcribed to cDNA with 200 U M-MLV-Reverse Transcriptase and 2.5 μM random hexamers. The cDNA samples (50 ng RNA equivalent in a 20 μL reaction volume) were subjected to PCR in duplicate on an I-Cycler (BioRad, Munich, Germany) using Maxima SYBR Green Mastermix (Thermo Fisher Scientific, Waltham, MA, USA). The threshold (C_t_) values of the genes-of-interest (GOI) were normalized to those of the housekeeping gene (HKG), TATA box-binding protein (TBP), in the same reaction using the 2-ΔΔCt method, and are displayed as relative mRNA expression. TBP was chosen for normalization since it is representative of an HKG with moderate-to-low expression (C_t_ values of ~25), being in the range of the C_t_ values (between 20 and 30) for most of the GOIs analyzed here. For a list of PCR primers see Appendix A.

### 5.3. Immunoblot Analysis

Cell lysis and immunoblotting was essentially performed as described previously [42]. In brief, cells were washed once with ice-cold PBS and lysed with PhosphoSafe lysis buffer (Merck Millipore, Darmstadt, Germany). Following clearance of the lysates by centrifugation, their total protein concentrations were determined with the DC Protein Assay (BioRad). Equal amounts of proteins were fractionated by PAGE on mini-PROTEAN TGX any-kD precast gels, or TGX Stain-Free FastCast gels (BioRad) and blotted to PVDF membranes. Membranes were blocked with nonfat dry milk or bovine serum albumin and incubated with primary antibodies (CgA: Monoclonal Mouse Anti-Human Chromogranin A, clone DAK-A3, Dako, Glostrup, Denmark; SYP: Anti-Synaptophysin, Dako; SSTR2: Anti-Somatostatin Receptor 2 antibody [UMB1]-C-terminal #ab134152, Abcam, Cambridge, UK) overnight at 4 °C. After washing and incubation with horseradish peroxidase-linked secondary antibodies, chemoluminescent detection of proteins was done on a ChemiDoc XRS imaging system (BioRad) with Amersham ECL Prime Detection Reagent (GE Healthcare, Munich, Germany). The signals for the proteins of interest were normalized to those for the housekeeping genes GAPDH or HSP90. 

### 5.4. Next Generation Sequencing of MicroRNAs

The isolation and purification of total RNA for next-generation sequencing (NGS) was done with Qiagen miRNeasy Mini Kit (Qiagen, Hilden, Germany). Total RNA was quality controlled on a TapeStation 4200 (Agilent, Waldbronn, Germany). RIN Scores were above 7 for all samples. A total of 200 ng of input total RNA were used to construct sequencing libraries using the NEXTFLEX® Small RNA-Seq Kit v3 for Illumina® Platforms (PerkinElmer, Waltham, MA, USA) following the manufacturers protocol for a gel-free workflow. Resulting libraries were quality controlled on a TapeStation 4200 (Agilent) and showed a clean peak at approximately 160 base pairs (bp). Quantification was performed on a Qubit 2.0 fluorometer (Thermo Fisher Scientific, location). The two libraries (corresponding to BON and QGP cells) were multiplexed and sequenced on a HiSeq 4000 (Illumina, San Diego, CA, USA) lane with 50 bp single-read sequencing.

Quality of raw fastq files was assessed using fastp v0.20.0 [69] (default options, percentage of unqualified bases limited to 30% and minimum read length was set to 30bp). Additionally, reads were trimmed from 5’ end 3’ by 5bp and adapter sequences (TGGAATTCTCGGGTGCCAAGG) were trimmed off. Trimmed reads mapped to GRCh38 using STAR aligner v2.7.2b [70] and Gencode v24 annotations. Additionally, miRNA annotations from Gencode V29 were added (ENCODE reference ENCSR564GPK). Resulting mapping files were filtered using R v3.6.1 (https://www.rdocumentation.org/packages/base).

### 5.5. Migration Assays

We employed the xCELLigence® DP system (ACEA Biosciences, San Diego, CA, USA, distributed by OLS, Bremen, Germany) to measure chemokinesis/-taxis of BON and QGP cells in real-time. Briefly, CIM plates-16 were prepared according to the instruction manual and previous descriptions [39]. The underside of the upper chambers of the CIM plate-16 was coated with 30 μl of collagen I or collagen IV to facilitate adherence of the cells and enhance signal intensities. After filling the wells of the lower and upper chambers of the CIM plate-16 with medium (containing 1% FBS in both the lower and upper chamber for the chemokinesis setup, or 10% FBS in the lower chamber and 0.5% FBS in the upper chamber for the chemotaxis setup), the plates were assembled and equilibrated in the incubator for 1 h. In both instances, 50,000–100,000 cells are applied to the upper chamber of each well. Data acquisition was done at intervals of 15 min and analyzed with RTCA software, version 1.2 (ACEA Biosciences).

### 5.6. Statistical Analysis

Statistical significance was calculated using the Mann–Whitney U test or the Wilcoxon test. Results were considered significant at *p* < 0.05 (indicated in the figures by asterisks).

## Figures and Tables

**Figure 1 cancers-12-00691-f001:**
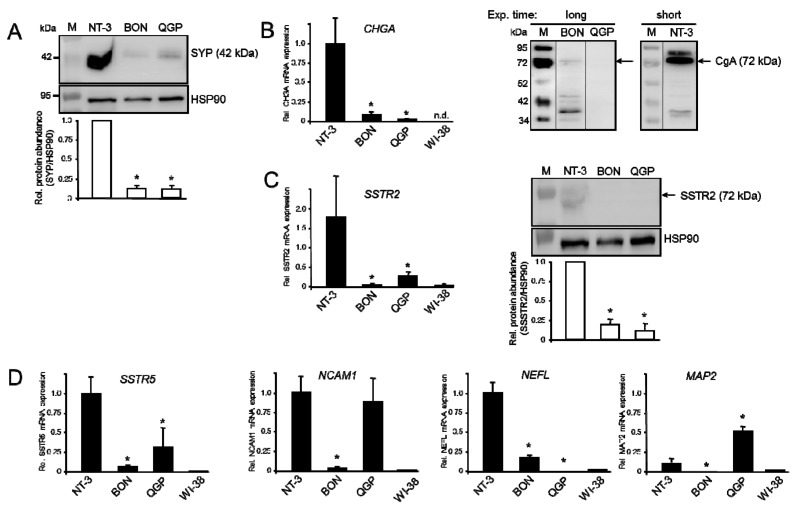
Expression of markers of neuroendocrine differentiation in BON and QGP cells, and NT-3 cells as control. (**A**) Immunoblot analysis of synaptophysin (SYP), and heat shock protein 90 (HSP90) as a loading control, in NT-3, BON, and QGP cells. The graph underneath the blot depicts the relative (Rel.) protein abundance calculated from the densitometric readings of band intensities (mean ± SD, *n* = 3) from three independent experiments, relative to NT-3 set arbitrarily at 1.0. The numbers to the left indicate band sizes of the molecular weight marker (M). (**B**) Quantitative real-time RT-PCR (qPCR, left-hand side) and qualitative immunoblot analysis (right-hand side) of chromogranin A (CgA). The qPCR data represent the mean ± SD from three to four cell preparations normalized to TATA box-binding protein (TBP). Signals for BON cells on immunoblots only became visible after an extended exposure (Exp.) time. The longer exposure time is also evident from the stronger bands of the molecular weight marker. The thin lines indicate removal of irrelevant lanes. (**C**) Immunoblot analysis of somatostatin receptor 2 (SSTR2) in NT-3, BON, and QGP cells. Results from densitometry-based quantification of the specific protein bands are given below the blots. (**D**) QPCR analysis of the indicated genes. Data were derived from three to four cell preparations. The asterisks (∗) indicate significance (*p* < 0.05, Wilcoxon test) relative to NT-3 cells. Human WI-38 fibroblasts were used as a negative control in some qPCR assays; *n*.d., not detectable.

**Figure 2 cancers-12-00691-f002:**
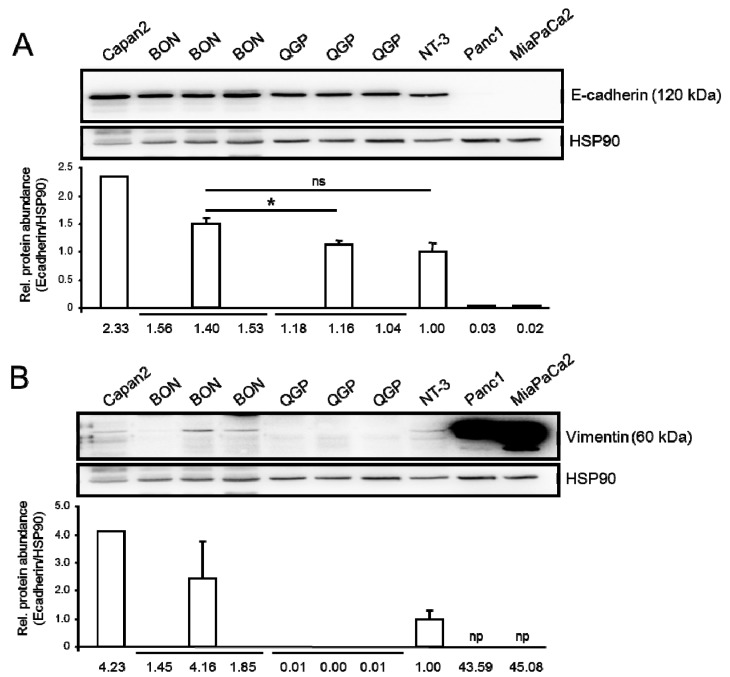
Expression of epithelial and mesenchymal markers in BON, QGP and NT-3 cells. (**A**) Immunoblot analysis of the epithelial marker E-cadherin. Capan2 or Panc1 and MiaPaCa2 cells were used as positive or negative control, respectively. (**B**) Immunoblot analysis of the mesenchymal marker Vimentin. Capan2 or Panc1 and MiaPaCa2 cells were used as negative or positive control, respectively. The graphs below the blots indicate densitometry-based quantification of band intensities (mean ± SD, *n* = 3) from three independent experiments (for NT-3 cells only the most representative one is shown). Data are plotted relative to NT-3 cells set arbitrarily at 1.0. The values for Panc1 and MiaPaCa2 were not plotted (np) as they do not fit the scale. The asterisk (∗) indicates a significant difference; ns, not significant.

**Figure 3 cancers-12-00691-f003:**
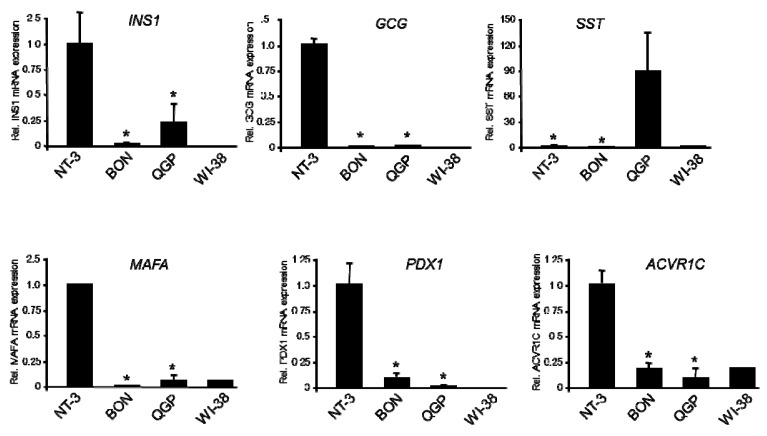
Quantitative real-time RT-PCR-based expression analysis of the indicated genes for islet hormones (INS1, GCG, SST) and marker genes of mature β-cell differentiation or function (MAFA, PDX1, ACVR1C) in BON, QGP, and NT-3 cells. Data represent the mean ± SD from three to four preparations. The asterisks (∗) indicate significant differences relative to NT-3 cells (*p* < 0.05, Wilcoxon test).

**Figure 4 cancers-12-00691-f004:**
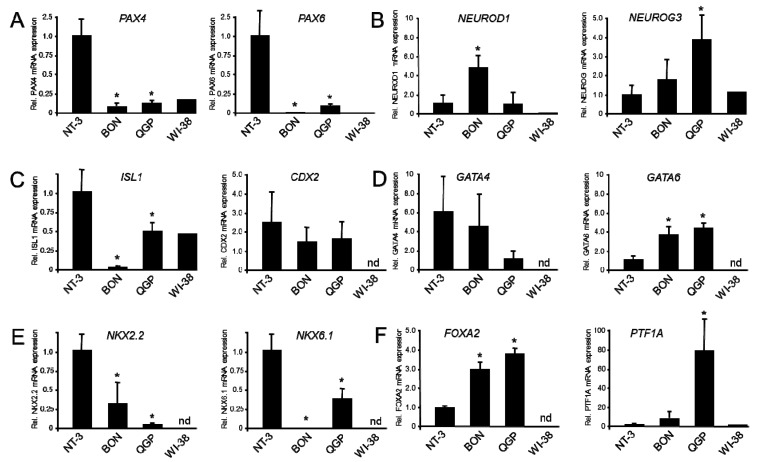
Quantitative real-time RT-PCR-based expression analysis of genes from immature β-cells and pancreatic (endocrine) progenitors in BON, QGP, and NT-3 cells. The indicated genes were grouped according to structural or functional similarities. (**A**) PAX4 and PAX6, (**B**) NEUROD1 and NEUROG3, (**C**) ISL1 and CDX2, (**D**) GATA4 and GATA6, (**E**) NKX2.2 and NKX6.1, (**F**) FOXA2 and PTF1A. Data represent the mean ± SD from three to four preparations. The asterisks (∗) indicate significant differences relative to NT-3 cells (*p* < 0.05, Wilcoxon test), nd, not determined.

**Figure 5 cancers-12-00691-f005:**
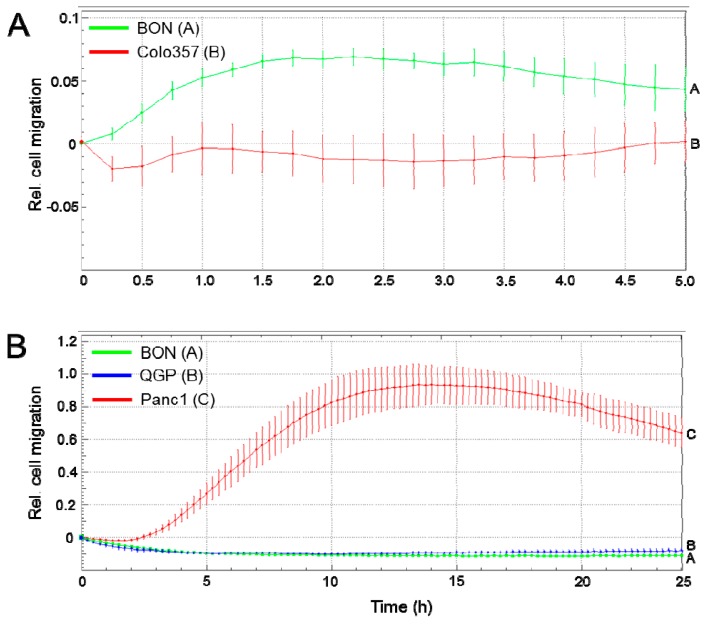
Cell migration assays of BON and QGP cells. (**A**) Assay in a chemokinesis setup with non-motile Colo357 cells as negative control. (**B**) Assay in a chemotaxis setup with Panc1 cells as positive control. Data represent the mean ± SD from four wells per condition. The assays shown are representative of three independent assays with very similar results.

**Figure 6 cancers-12-00691-f006:**
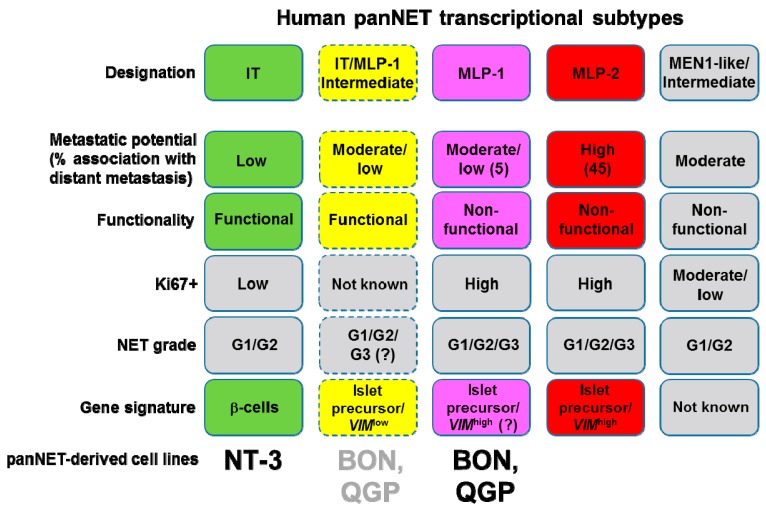
Cartoon showing allocation of NT-3, BON, and QGP cells to the transcriptional subtypes of pancreatic NET (panNET) described in [12]. Based on the data from phenotypic characterization and gene expression analyses, NT-3 cells can be classified as IT, while BON and QGP cells more closely resemble the MLP subtype. Due to the well-differentiated epithelial phenotype, the expression of miRs with anti-EMT and anti-metastatic function, and the low or absent migratory activity in vitro, BON and QGP cells have been tentatively grouped into the MLP-1 sub-subtype of tumors. In case of EMT marker (i.e., *VIM*), expression in the MLP-1 sub-subtype can be shown to be as high as in the MLP subtype (combining MLP-1 and MLP-2), then BON and QGP cells more likely belong to a still hypothetical subtype that combines features of both MLP and IT tumors (designated IT/MLP-1 Intermediate, yellow boxes with stippled lines). Tumors of this class are assumed to be still functional with respect to serotonin secretion of BON and SST secretion of QGP cells. Adapted from [12], modified.

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
