# Peer review of "A Comprehensive Molecular Characterization of the Pancreatic Neuroendocrine Tumor Cell Lines BON-1 and QGP-1"

_cancers, 2020, doi:10.3390/cancers12030691_

Round 1

Reviewer 1 Report

The manuscript is aimed at performing a wide screening of molecular features of pNEN CLs. Overall, the results are interesting and the experiments seem soundly performed.

My main criticisim regards the fact that the paper looks more like a long list of experiments performed separately on these cells rather than like a meaningful, hypothesis-driven, effort to investigate them.

there are too many data, a lot of the results are discusssed within the results section and this makes it very unpleaseant to be read.

To me all the bit on Mir seems unesufulor at leastthe autors should try and put the results in a more logic way, discussing hypothesis regarding cell origin, hints for response to treatments and so, instead of giving a ver ylong list.

For example:

As for the data on cell migration, the authors should discuss results of studies investigating migratory capacity of these cells on different scaffolds depending on treatments (cfr. PMID: 21712346).

Similarly, finding of expresison of MiR-18a in QGP might have to do with the poor sensitivity of these cells to rapalogs (PMID: 25026292).

The authors mention the CL CM at the beginning; it could ne nice to see some data on it.

Author Response

Dear Editor, dear Mrs. Zhang:

First of all, we sincerely thank the reviewers for their enthusiatic comments on our manuscript. We have done our best to incorporate their suggestions in the revised version. Changes and additions to the original version have been highlighted in the “track changes” mode. We hope that the Reviewers are satisfied with the revised version and that they consider our manuscript suitable now for publication.

Kind regards,

Hendrik Ungefroren

The manuscript is aimed at performing a wide screening of molecular features of pNEN CLs. Overall, the results are interesting and the experiments seem soundly performed.

Response: We thank the reviewer for this positive comment.

  1. My main criticism regards the fact that the paper looks more like a long list of experiments performed separately on these cells rather than like a meaningful, hypothesis-driven, effort to investigate them.

Response: We have, whereever possible, presented a hypothesis, i.e. in the last paragraph of the Introduction and at the beginning of the sections 2.3. (lines 198-201), 2.5 (second paragraph, lines 302-305) and 2.6. (lines 328-329). In sections 2.2. and 2.4. the respective hypotheses had already been formulated in the original version.

  1. There are too many data, a lot of the results are discussed within the results section and this makes it very unpleasant to be read.

Response: We did not want to reduce the number of data presented, however, we agree with the reviewer that extensively discussing the data in the Results section impairs reading fluency. For this reason, we have removed at several occasions sentences with data interpretation from the Results section, either by complete deletion or by moving them to the Discussion section.

  1. To me all the bit on Mir seems unuseful or at least the authors should try and put the results in a more logic way, discussing hypothesis regarding cell origin, hints for response to treatments and so, instead of giving a very long list. For example: As for the data on cell migration, the authors should discuss results of studies investigating migratory capacity of these cells on different scaffolds depending on treatments (cfr. PMID: 21712346).

Response: As requested, we have arranged the miR data in a more structured and logic way based on hypotheses regarding cellular function, role in oncogenesis and response to treatment (see lines 302-305 in the revised version). The discussion on some miRs that are unrelated to these functions has been removed.

As suggested, we have discussed the implications of a study reporting that treatment with everolimus increased the activity of Src which might result in an enhanced migratory capacity of BON and QGP cells on different scaffolds (lines 434-437). Two additional references (PMID: 21712346 and PMID: 17395980) related to this issue have been added to the Reference list (#54 und 55).

  1. Similarly, finding of expression of MiR-18a in QGP might have to do with the poor sensitivity of these cells to rapalogs (PMID: 25026292).

Response: We thank the reviewer for providing this piece of information. We have included this and the corresponding reference (#52) in the Discussion.

  1. The authors mention the CL CM at the beginning; it could be nice to see some data on it.

Response: Unfortunately, we do not readily have access to this cell line. A literature search revelead that there is some dispute as to whether this cell line is a suitable model for the study of β-cell function in vitro due to chromosomal abnormalities, inconsistent insulin secretion/content, its polyclonal nature and fibroblast-like morphology [Jonnakuty & Gragnoli. Karyotype of the human insulinoma CM cell line--beta cell model in vitro? J. Cell. Physiol. 2007, 213, 661-2; Gragnoli. The CM cell line derived from liver metastasis of malignant human insulinoma is not a valid beta cell model for in vitro studies. J. Cell. Physiol. 2008, 216, 569-70]. For these reasons we believe that adding data with this cell line will not strengthen our conclusions.

Reviewer 2 Report

This is a manuscript by Luley KB et al about BON-1 and QCP-1 NET cell lines and their molecular characterization. This is a well-designed study with comprehensive amount of laboratory work. The results help researchers to decide about which cell lines are appropriate for specific experiments.

Minor points:

I suggest that the discussion about BON1/QGP and CgA/insulin pathway will be adjusted. CgA is not evident in b cells (eg The Journal of Histochemistry & Cytochemistry, Volume 49(4): 483–490, 2001) and according to the literature is not an effective biochemical marker for insulinomas (eg Qiao et al. BMC Endocrine Disorders 2014, 14:64).

The discussion is long and some effort to make it shorter would be highly appreciated.

Author Response

Dear Editor, dear Mrs. Zhang:

First of all, we sincerely thank the reviewers for their enthusiatic comments on our manuscript. We have done our best to incorporate their suggestions in the revised version. Changes and additions to the original version have been highlighted in the “track changes” mode. We hope that the Reviewers are satisfied with the revised version and that they consider our manuscript suitable now for publication.

Kind regards,

Hendrik Ungefroren

This is a manuscript by Luley KB et al about BON-1 and QCP-1 NET cell lines and their molecular characterization. This is a well-designed study with comprehensive amount of laboratory work. The results help researchers to decide about which cell lines are appropriate for specific experiments.

Response: We thank the reviewer for this enthusiastic comment.

Minor points:

  1. I suggest that the discussion about BON1/QGP and CgA/insulin pathway will be adjusted. CgA is not evident in b cells (eg The Journal of Histochemistry & Cytochemistry, Volume 49(4): 483–490, 2001) and according to the literature is not an effective biochemical marker for insulinomas (eg Qiao et al. BMC Endocrine Disorders 2014, 14:64).

Response: As requested, we have discussed these important aspects in the third paragraph of the Discussion section (including new ref. #42) and have removed and rephrased the sentence in line 384.

  1. The discussion is long and some effort to make it shorter would be highly appreciated.

Response: As desired, we have slightly shortened the Discussion.